# Pasteurian Contributions to the Study of *Bordetella pertussis* Toxins

**DOI:** 10.3390/toxins15030176

**Published:** 2023-02-25

**Authors:** Camille Locht

**Affiliations:** Univ. Lille, CNRS, Inserm, CHU Lille, Institut Pasteur de Lille, U1019-UMR 9017-CIIL-Centre for Infection and Immunity of Lille, F-59000 Lille, France; camille.locht@pasteur-lille.fr

**Keywords:** *Bordetella*, lipo-oligosaccharide, adenylyl cyclase toxin, pertussis toxin, vaccines

## Abstract

As a tribute to Louis Pasteur on the occasion of the 200th anniversary of his birth, this article summarizes the main contributions of scientists from Pasteur Institutes to the current knowledge of toxins produced by *Bordetella pertussis*. The article therefore focuses on publications authored by researchers from Pasteur Institutes and is not intended as a systematic review of *B. pertussis* toxins. Besides identifying *B. pertussis* as the causative agent of whooping cough, Pasteurians have made several major contributions with respect to the structure–function relationship of the *Bordetella* lipo-oligosaccharide, adenylyl cyclase toxin and pertussis toxin. In addition to contributing to the understanding of these toxins’ mechanisms at the molecular and cellular levels and their role in pathogenesis, scientists at Pasteur Institutes have also exploited potential applications of the gathered knowledge of these toxins. These applications range from the development of novel tools to study protein–protein interactions over the design of novel antigen delivery tools, such as prophylactic or therapeutic vaccine candidates against cancer and viral infection, to the development of a live attenuated nasal pertussis vaccine. This scientific journey from basic science to applications in the field of human health matches perfectly with the overall scientific objectives outlined by Louis Pasteur himself.

## 1. Introduction

The *Bordetella* genus contains over a dozen different species, four of which are bona fide pathogens for warm-blooded vertebrates [1]. *Bordetella pertussis* causes whooping cough in humans. The human-adapted lineage of the ovine pathogen *Bordetella parapertussis* cases mild whooping-cough-like symptoms in humans. *Bordetella bronchispetica* is able to infect many mammals and causes rhinitis in pigs and kennel cough in dogs. It may also infect humans and is considered an opportunistic pathogen for humans, as it can cause symptomatic infections in individuals with immune deficiency. Finally, *Bordetella avium* is a bird pathogen. Other *Bordetella* species, such as *Bordetella holmesii*, have also occasionally been associated with human disease, although mostly in immunocompromised individuals. In this article, I briefly review the contribution of scientists from Pasteur Institutes to the understanding of *Bordetella* toxins.

## 2. From the Identification of the Whooping Cough Agent to the Concept of *Bordetella* Toxins

The first *Bordetella* species that was discovered is *B. pertussis*, initially called *Haemophilus pertussis*, because of a haemolytic halo that can be observed when the organism is grown on blood agar plates, suggesting the presence of a haemolysin toxin. *B. pertussis* was first observed by Jules Bordet in 1900 [2] when he was at the Pasteur Institute in Paris and examined the expectoration of his 5-month-old daughter, who was suffering from typical whooping cough, under a microscope. However, the organism was cultured for the first time only in 1906 and definitively identified as the whopping cough agent by Jules Bordet, at the Institut Pasteur de Bruxelles, together with his brother-in-law, Octave Gengou. It became clear very quickly that successful culturing of *B. pertussis* only occurred very early after the onset of the disease, while the symptoms last for weeks to months after the organism has disappeared. This observation led Bordet and Gengou to postulate the action of toxic compounds and to qualify pertussis as a toxin-mediated disease. The presence of toxic substances was further confirmed by the observation that the intraperitoneal injection of killed *B. pertussis* extracts led to a pleural effusion and eventually the death of guinea pigs [3]. 

## 3. Lipo-Oligosaccharide

The first evidence of a *Bordetella* toxin, initially named endotoxin, was presented by Bordet and Gengou in a publication in 1909 [4] that referred to a toxin present in whole-cell lysates with strong toxicity in rabbits and guinea pigs. When injected into mice, rabbits or guinea pigs, it induced dermonecrotic lesions and death of the animals. Toxicity was lost upon heating for 30 min at 56 °C. Because of its heat lability, it is likely that this toxin corresponds to what has later been referred to as heat-labile or dermonecrotic toxin, a 1464 amino-acid-long protein toxin [5]. 

However, in addition to the heat-labile dermonecrotic toxin, *Bordetella* species also produce a non-proteinaceous endotoxin, which corresponds to the *Bordetella* lipo-oligosaccharide (LOS). This molecule was extensively characterized by Martine Caroff and colleagues [6], and its action on immune cells was investigated by Jean-Marc Cavaillon and colleagues at the Institut Pasteur in Paris. His team studied the structure–function relationship of *Bordetella* LOS and found a specific binding of this molecule to lectin-like receptors on the surface of macrophages [7] and subsequent polyclonal B-cell activation. The polysaccharide moiety of the *Bordetella* LOS further stimulated IL-1 secretion by macrophages and monocytes [8]. The precise LOS structural determinant responsible for the induction of IL-1 secretion was identified as the inner core region consisting of 2-keto-3-deoxy-D-manno-octulosonic acid and heptose [9]. In contrast, the lipid A moiety of the *Bordetella* LOS was found to be 1000 to 10,000 times less effective in inducing IL-1 secretion than the complete LOS molecule [10]. 

## 4. Adenylyl Cyclase/Haemolysin Toxin

### 4.1. Biogenesis

As previously observed by Bordet and Gengou, growth of *B. pertussis* on blood agar plates results in the appearance of a haemolytic halo surrounding the bacterial colonies, suggesting the production and secretion of a haemolytic molecule. This haemolytic activity is carried by the C-terminal domain of a 177.312 kDa protein, which contains an N-terminal catalytic domain expressing potent adenylyl cyclase activity [11]. A highly potent secreted adenylyl cyclase was first identified by Hewlett et al. [12], but the conclusive link of this enzyme with the haemolysin activity of *Bordetella* came from the seminal discovery of its corresponding structural gene by elegant work performed at the Institut Pasteur in Paris by Glaser et al. [13]. It was known that the *Bordetella* adenylyl cyclase needed calmodulin to express its enzyme activity. By generating an *Eschericia coli* strain that overproduced recombinant calmodulin, the teams at the Institut Pasteur identified *B. pertussis* DNA fragments able to complement adenylyl cyclase deficiency in this *E. coli* strain. The isolation and sequence of the full-length gene encoding the adenylyl cyclase, named *cyaA*, revealed an open reading frame of 1706 codons and predicted a protein consisting of four domains, including the N-terminal 400-residue-long calmodulin-sensitive cyclase domain. The C-terminal domain was hypothesized to possess haemolytic properties, which was confirmed by subsequent work by the same team [14]. They uncovered the similarities between this C-terminal domain and the *E. coli* alpha-haemolysin and with the *Pasteurella haemolytica* leukotoxin. Furthermore, they established the parallelism between the *E. coli* alpha-haemolysin secretion mechanism and the secretion mechanism of the *Bordetella* adenylyl cyclase/haemolysin toxin (ACT) by sequencing the *cyaA* downstream region and identifying two genes, named *cyaB* and *cyaD*. These two genes code for a 712- and a 400-residue-long protein, which are similar to *E. coli* HlyB and HlyD, respectively, both of which are involved in the secretion of the *E. coli* alpha-haemolysin. However, the *Bordetella* ACT secretion also required an additional protein, the gene of which was identified downstream of *cyaD* and named *cyaE*. The production of *Bordetella* ACT and its secretion as a single 200 kDa polypeptide chain was confirmed by Bellalou et al. in subsequent studies [15]. 

In addition to the identification of adenylyl cyclase and haemolysin as a single protein and its secretion apparatus, the team also made important contributions to the understanding of the regulation of its expression. The expression of the *Bordetella* virulence genes is under the control of the two-component BvgA/S signal-transducing system. BvgA is a transcriptional activator, and BvgS is a sensor protein that transmits information on environmental changes to BvgA via a phosphorylation cascade [16]. Laoide and Ullmann [17] found that the *cyaA* gene is under the control of BvgA, while transcription of the downstream *cyaBDE* cistrons, which are separated by 77 nucleotides from the *cyaA* translational stop codon, is independent of BvgA. Using primer-extension analyses, they identified the transcriptional start site of *cyaA* and observed transcription only under BvgA^+^ conditions. Using the same technology, they also identified the transcriptional start site of the *cyaBDE* operon and found that this operon was transcribed under both BvgA^+^ and BvgA^−^ conditions. However, the strength of the latter promoter was 4 to 5 times lower than that of the *cyaA* gene. Steffen et al. [18] then demonstrated that, in contrast to some other BvgA-regulated genes, the *cyaA* gene requires phosphorylation of BvgA in order to be activated by this transcription factor and that phosphorylation of BvgA is sufficient to activate transcription of *cyaA*. This was demonstrated by the development of an in vitro transcription system using the RNA polymerase of *Bordetella*. In a series of runoff transcription experiments, they showed that the *Bordetella* RNA polymerase was able to efficiently drive transcription of the *cyaA* gene in the presence of phosphorylated BvgA. When *E. coli* RNA polymerase was used instead of the *Bordetella* enzyme, transcription levels were markedly reduced, suggesting that the *E. coli* RNA polymerase is less efficient in the formation of the transcription initiation complex than its *Bordetella* counterpart. 

### 4.2. Calmodulin Binding and Catalysis

The structure–function relationship of ACT was also extensively studied by scientists of the Institut Pasteur in Paris. Considering the importance of calmodulin binding for adenylyl cyclase activity, Ladant examined the interaction of the enzyme with calmodulin and found that the two molecules interact with each other in a 1:1 stoichiometry and that the interaction was stronger in the presence of calcium than in its absence [19]. Trypsin treatment of calmodulin-bound adenylyl cyclase resulted in two distinct fragments of 18 and 25 kDa, respectively, both of which interacted with calmodulin. After trypsin cleavage, the two fragments remained catalytically active when bound to calmodulin, whereas in the absence of calmodulin, the two fragments lost their catalytic activity. The 18-kDa fragment was then identified as the main calmodulin-binding domain, and the 25-kDa fragment located at the N-terminal moiety was identified as the catalytic domain [20]. Using a fluorescent ATP analogue, Sarfati et al. [21] demonstrated that in the presence of calcium, calmodulin increased substrate binding of the adenylyl cyclase, probably through the induction of conformational changes. Work by Bouhss et al. [22] narrowed down the essential peptide for calmodulin binding to a 72 amino acid peptide that spans the C terminus of the 25-kDa peptide and the N terminus of the 18-kDa peptide and showed that the hydrophobic helix around Trp-242 is critical for calmodulin binding. However, the N-terminal half of the catalytic 25-kDa domain also contributes to calmodulin binding, albeit to a much lesser degree. Regions critical for calmodulin binding were subsequently identified around Leu-247 and Cys-335 by Ladant et al. [23] using an insertional mutagenesis approach. Further mutagenesis experiments using an alanine substitution approach revealed that replacement of Arg-338, Asn-347 and Asp-360 by alanine dramatically reduced the affinity of adenylyl cyclase to calmodulin, and molecular dynamics simulations suggested that the mutations may have caused large fluctuations of the calcium-binding loops of calmodulin, which may have weakened the interaction of the enzyme with calmodulin and destabilized the catalytic loop of the adenylyl cyclase [24]. More recently, O’Brien et al. [25] reported that calmodulin helps the adenylyl cyclase enzyme to adopt the folding necessary to express its enzymatic activity, as evidenced by a series of small-angle X-ray scattering, hydrogen/deuterium exchange mass spectrometry and synchrotron radiation circular dichroism measurements. In the absence of calmodulin, adenylyl cyclase is structurally disordered, with a 75-residue-long peptide in the 18-kDa C-terminal domain, spanning residues 201 to 275. Binding of calmodulin to this region transitions the domain from a disordered to an ordered conformation and induces allosteric effects to stabilize the distant catalytic site. This intrinsically disordered region may destabilize the enzyme in the absence of calmodulin so that it is only active when needed, i.e., within the target cell where calmodulin is present. The intrinsically disordered structure may also be helpful for the toxin to cross the bacterial secretion system and the target cell membrane.

The catalytic mechanism was investigated by Glaser et al. [26], first by using site-directed mutagenesis to identify critical amino acid residues for enzyme function. Replacement of lysines at position 58 or 65 by glycines strongly reduced the catalytic activity. Later, four additional residues, i.e., Asp-188, Asp-190, His-298 and Glu-301, were found to be important for enzyme activity [27]. The substitution of these amino acids had a relatively minor impact on calmodulin binding. Instead, binding to a photoactivable ATP analogue was strongly affected, especially by substitutions of Asp-190. A mechanism was proposed whereby Asp-188 and Asp-190 of the enzyme interact with Mg^++^-ATP, thereby stabilizing the transition state. In this model, Lys-65 or Lys-58 interacts with the α-phosphate group of the Mg^++^-ATP. Another amino acid can then act as a basic catalyst for the cyclization of the ATP. 

Another important residue for catalysis is His-63 [28]. While replacement of His-298 by leucine or arginine did strongly reduce enzyme activity, it did not alter the kinetic characteristics of the adenylyl cyclase. In contrast, substitution of His-63 altered its kinetic properties, indicating that this residue is directly involved in catalysis itself, possibly as part of a charge relay system in a general acid/base enzyme mechanism. 

### 4.3. Haemolytic and Receptor-Binding Domain

As shown by Glaser et al. [14] and Bellalou et al. [15], the haemolytic activity of ACT is confined to the 1306 C-terminal residues of the molecule, showing sequence similarities to the *E. coli* α-haemolysin, although its haemolytic activity is weaker than that of the *E. coli* α-haemolysin. By constructing mutant derivatives of ACT devoid of adenylyl cyclase activity or lacking the catalytic N-terminal domain, Sakamoto et al. [29] showed that haemolysis does not require enzyme activity and that the C-terminal moiety of the molecule is sufficient to express full haemolytic activity. 

The latter domain is also essential for the transport of the catalytic domain into the target cell and is itself composed of two distinct subdomains. The N-proximal subdomain contains hydrophobic segments, which are essential for haemolytic and toxin activity [15]. By using an artificial planar–lipid bilayer system, Benz et al. [30] demonstrated that ACT forms small ion-permeable pores in lipid bilayers with a very small diameter dependent on the subdomain containing the hydrophobic segments. However, the pore-forming activity of the native protein was substantially higher than that of a recombinant analogue, suggesting the role of a post-translational modification of the toxin. 

The post-translational modification of ACT is catalysed by the product of an additional gene, namely *cyaC*, which was discovered by Barry et al. [31] at the Medical College of Virginia. This allowed Sebo et al. [32] to produce fully active ACT by recombinant *E. coli* co-expressing *cyaC*, together with *cyaA* [32]. Through in vitro chemical acylation of lysines on non-modified, inactive, recombinant ACT, Heveker et al. [33] activated ACT by transferring lauric, myristic or palmitic acid chains, as evaluated by both its haemolytic and toxic activities. However, the activity of the chemically modified ACT was still lower than that of native ACT. Nevertheless, this study suggests that activation of ACT by CyaC occurs through acylation of one or more lysine residue(s) of the toxin. The modified lysine residues were identified as Lys-983 and, potentially, Lys-860 by the Hewlett group at the University of Virginia [34,35] and found to be palmitoylated by CyaC. 

Initially, it was hypothesized that the domain containing the hydrophobic segments would form channels in the target cell membranes through which the catalytic domain can gain access to the cytoplasm. However, the size of the channels is too small to allow for translocation of a protein domain, even when totally unfolded [30]. Karst et al. [36] identified an additional segment spanning residues 375–485 located at the C-terminal end of the catalytic domain that can also bind to membranes and is able to destabilize lipid bilayers. When this segment was deleted from ACT, toxin activity was abrogated. This segment contains two subsegments spanning residues 414 to 440 and 454 to 484, respectively, that are predicted to form α-helical structures with membrane-interacting potential [37]. A synthetic peptide corresponding to one of these two subsegments was shown to bind to and permeabilize membranes containing anionic lipids. It adopted an α-helical conformation upon interaction with the lipid bilayer and was able to permeabilize lipid vesicles. For the peptide to acquire its secondary structure, the anionic nature of the lipids is important, most likely via the interaction of the lipid with the two positively charged arginine residues of the peptide. More recently, Voegele et al. found that this peptide can translocate across lipid bilayers and then bind to calcium-loaded calmodulin, thereby pulling the entire catalytic domain through the plasma membrane [38]. This interaction with calmodulin appears to be essential for translocation, as substitutions of amino acid residues that interact with calmodulin or the use of calmodulin inhibitors abrogates translocation. These observations indicate that calmodulin plays a double role in ACT action by aiding in membrane translocation of the adenylyl cyclase domain and by stabilizing the active conformation of the enzyme. 

Calcium plays an important role in ACT action not only by binding to calmodulin. Both ACT toxicity and haemolysis are calcium-dependent [39], and the toxin can directly bind calcium via several high-affinity and low-affinity binding sites [40]. Binding of calcium to high-affinity sites was proposed to be necessary for the haemolytic activity of the toxin, while binding to the low-affinity sites induces conformational rearrangements of the protein. The latter sites were mapped to the C-terminal part of the protein characterized by Asp-Gly-rich repeats common to the members of the RTX (for repeat in toxin) family. Work by Bauche et al. [41] established that the calcium-binding sites extend beyond the Asp-Gly-rich repeats and include adjacent protein segments. These adjacent segments are essential for the calcium-binding sites to fold into a stable parallel ß helix upon binding to calcium. In the absence of calcium, the Asp-Gly-rich repeat region is intrinsically disordered [42]. Since the calcium concentration is low within the bacterial cytosol, the newly synthesized ACT is likely in an unfolded conformation, which was shown to be important for ACT to be secreted via a type 1 secretion system through the *B. pertussis* cell wall. Once secreted, ACT can bind calcium and thereby adopt its functional conformation. The disorder-to-order transition triggered by calcium binding also reduces the mean net charge of ACT, which influences its ability to multimerize [43]. This calcium effect strongly depends on the C-terminal flanking region of the Asp-Gly repeats. Like many other RTX proteins, ACT tends to form multimers in vitro, such as after a denaturing/renaturing cycle of purified ACT. However, the monomeric form displays substantially stronger haemolytic and toxin activity than the multimeric forms [44], suggesting that the physiologically active form of ACT is monomeric, which is a form favoured by calcium and post-translational acylation. Nevertheless, the fact that membrane permeabilization by ACT requires oligomerization suggests that the protein may oligomerize once it has partitioned into the membrane. 

The Asp-Gly-rich C-terminal region of ACT is also its receptor-binding domain. The ACT receptor was identified by Guermonprez et al. [45] as the α_M_β_2_ (CD11b/CD18) integrin expressed on the surface of leukocytes, as evidenced by the ability of anti-CD11b and anti-CD18 antibodies (but not by antibodies to other integrins) to block ACT binding to neutrophils and ensuing increases in intracellular cAMP levels. Although ACT can bind to and invade many different cell types, Chinese hamster ovary cells became highly sensitive to ACT when transfected with CD11b/CD18 but not with other integrins, and binding to CD11b/CD18 strictly depended on the presence of calcium. CD11b/CD18 is mostly expressed on the surface of macrophages, neutrophils and dendritic cells, suggesting a primary action of ACT on these innate immune cells. 

### 4.4. Biological Activities of ACT

The team headed by Nicole Guiso has extensively studied the biological functions of ACT. Thirty years ago, this group showed that macrophages are indeed a major cell type targeted by ACT [46], which is consistent with the presence of the ACT receptor in these cells. Macrophages infected with *B. pertussis* undergo apoptosis, which depends on ACT, as infection with ACT-deficient *B. pertussis* does not induce apoptosis. Both the catalytic and the haemolytic domains are required for apoptosis induction. In vivo, in a murine respiratory challenge model, infection by *B. pertussis* was shown to induce apoptosis of macrophages, along with neutrophils, present in bronchoalveolar lavage fluids and lung tissues [47]. In vivo macrophage apoptosis was not observed when mice were infected with ACT-deficient *B. pertussis* strains, while neutrophil apoptosis was still observed in these mice. ACT-deficient *B. pertussis* strains were also affected in their growth rate during the early phase of infection in a murine respiratory challenge model [48], and both the haemolytic and the catalytic domains were shown to be important for early growth during infection. Furthermore, ACT was found to co-operate with pertussis toxin (PTX) in the infectious process and in the induction of histopathological lesions, as well as in the recruitment of inflammatory cells, including neutrophils, in the lungs [47,49]. 

In addition to apoptosis of macrophages, ACT also induces more subtle responses of monocytes. The increase in intracellular cAMP levels within human monocytes upon *B. pertussis* infection is inversely correlated with TNF-α production, superoxide anion release and Hsp70 expression [50]. These effects depend on the production of ACT and were not observed when the cells were incubated with an ACT-deficient strain but could be reproduced by incubation in the presence of purified ACT, suggesting that the toxin affects monocyte responses to infection to evade innate antimicrobial defence strategies. 

ACT may also play a role in the interaction of *B. pertussis* with epithelial cells. Curiously, ACT-deficient mutants were able to invade human HTE tracheal epithelial cells more efficiently than fully virulent isogenic strains [51], suggesting that ACT prevents invasion of these cells. *B. pertussis* also induces the secretion of IL-6 by HTE cells in an ACT-dependent manner [52], indicating its role in the inflammatory process triggered by *B. pertussis* infection. 

The protective effect of ACT as a potential vaccine antigen against pertussis was also tested by the Guiso team using a mouse intranasal infection model. Administration of anti-ACT antibodies to 3-week-old female BALB/c mice significantly protected them from pulmonary and intracerebral lesions, as well as death induced by challenge with high doses of the 18,323 strain of *B. pertussis*, as did active immunization with the purified catalytic domain of ACT [53]. Follow-up studies showed that ACT and its catalytic domain protected mice equally well against bacterial lung colonization in a sublethal model of *B. pertussis* infection, albeit not as well as a whole-cell vaccine comparator [54] due to neutralizing antibodies binding to residues 373–400 of the catalytic domain that block toxin translocation. However, later studies by the same group showed that the C-terminal domain of ACT is essential for protection against *B. pertussis* colonization. Using a series of truncated ACT derivatives, Betsou et al. [55] found that only immunization with derivatives that contained the Asp-Gly-rich repeat region conferred protection against lung colonization by *B. pertussis*. This region is also the predominant part of the protein that elicits antibodies upon infection by *B. pertussis* in mice, as well as in humans. In addition to this region, the last 217 amino acids of ACT are also essential for antibody recognition and protective activity, most likely because of its role in folding, e.g., to display the protective epitopes. ACT derivatives that lacked the Asp-Gly-rich repeat region were nevertheless able to induce strong ACT-neutralizing antibodies, suggesting that protection induced by ACT may be independent of elicited neutralizing antibodies. CyaC-catalyzed acylation was also found to be important for ACT-mediated protection against *B. pertussis* lung colonization [56]. This was shown by comparing recombinant ACT produced by *E. coli* in the presence or absence of coexpressed *cyaC*. In a sublethal nasal challenge model, recombinant ACT produced in the absence of coexpressed *cyaC* provided no protection at all against lung colonization. In contrast, when the mice were immunized with CyaC-activated ACT, a significant reduction in bacterial colonization of the lungs was observed compared to non-vaccinated mice. Partial activation through low levels of *cyaC* coexpression resulted in intermediate levels of protection. However, ACT produced in *cyaC*-co-expressing *E. coli* was less protective than native ACT, possibly due to differences in the chemical nature of the acyl group between the recombinant and the native forms of ACT or to the presence of small amounts of contaminating antigens in the native ACT preparation. 

### 4.5. Applications

The various unique properties of ACT have prompted the proposal of potential applications in a variety of different fields. Its invasive properties led Sebo et al. [57] to propose ACT as a vehicle to deliver foreign epitopes directly into the cytosol for the induction of CD8^+^ cytotoxic T cells. When target cells were invaded by a recombinant ACT analogue that carries a CD8^+^ T-cell epitope from the lymphocytic choriomeningitis virus (LCMV), they were found to be readily lysed by cytotoxic CD8^+^ T cells specific to this epitope, indicating that ACT may be a good carrier for T-cell epitopes to be presented by the major histocompatibility complex class I molecules. Moreover, active immunization with enzymatically inactive hybrid ACT molecules carrying viral epitopes of LCMV or human immunodeficiency virus (HIV) resulted in antigen-specific class I restricted CD8^+^ CTL responses that could kill peptide-loaded target cells [58]. Using a mouse model of lethal intracerebral LCMV infection, Saron et al. [59] found that genetically inactivated hybrid ACT molecules carrying the LCMV epitope protected the animals from death through a CD8^+^ T-cell-dependent mechanism. The use of hybrid ACT analogues for the presentation of epitopes to CD8^+^ T cells was then extended to other antigens, including tumour antigens [60], which led to the finding that net negative charges are detrimental to the translocation process and the ensuing induction of CD8^+^ T-cell responses. Furthermore, proof of concept was provided that this approach can stimulate antitumour immunity [61]. The ACT vector was further improved by inserting multiple copies of MHC class I and class II restricted epitopes, which, in addition to CD8^+^ CTL responses, triggered potent Th1-type immune responses, as evidenced by high production of IL-2 and IFN-γ [62]. The ACT vector was also found to be capable of accommodating several different CD8^+^ T-cell epitopes, as exemplified by an LCMV polyepitope, an HIV and an ovalbumin epitope inserted simultaneously at three different permissive sites of ACT [63]. After administration to mice, all three were processed and induced CTL responses, and the immunized mice were protected against lethal challenge with LCMV. Importantly, prior immunization with ACT did not appear to prevent the induction of CD8^+^ T-cell responses to the grafted epitopes by the recombinant ACT analogues. 

Given that ACT preferentially binds to CD11b/CD18-expressing cells, including dendritic cells, targeting foreign antigens to these antigen-presenting cells may be an efficient way to not only induce CD8^+^ CTL responses but also T-helper cell and B-cell responses. This was shown by using a recombinant ACT derivative carrying the full-length HIV-1 Tat protein [64]. Immunization with this hybrid toxin in the absence of adjuvant induced high levels of long-lasting Tat-specific neutralizing antibodies and Th1-type T-cell responses. The same molecule was also able to induce strong neutralizing antibody and Th1-type T-cell responses in African green monkeys [65], indicating that ACT can also deliver foreign antigens to dendritic cells of primates. 

Similarly, full-length and various segments of the E7 oncoprotein from the human papilloma virus 16 were inserted into genetically detoxified ACT and were shown to elicit strong CTL and Th1 responses in mice [66]. The vaccines were also able to trigger tumour regression in mice injected with E7-expressing tumour cells. Tumour regression reached up to 100%, accompanied by 100% survival, which exceeded that of the comparator consisting of the E7 peptide administered together with CpG ODN 1826. This vaccine candidate, combined with a HPV18 E7-ACT protein, is now in clinical development and was shown to be safe and immunogenic in women infected with HPV16 or HPV18 [67]. 

The observation that the catalytic domain of ACT can be split into two fragments of 18 and 25 kDa, respectively, that can complement each other in the presence of calmodulin [19,20] has led to the development of a two-hybrid system to study protein–protein interactions [68]. Separately, the two fragments do not express adenylyl cyclase activity in the absence of calmodulin. However, when two proteins that interact with each other are fused to the two complementary 18-kDa and 25-kDa ACT fragments, enzyme activity is restored, and high levels of cAMP are produced, even in the absence of calmodulin. cAMP can then bind to the transcriptional activator CAP, which triggers the transcription of catabolic operons, including the lactose and maltose operons. This can easily be monitored phenotypically in an *E. coli* strain that lacks the endogenous adenylyl cyclase by the use of indicator plates. This bacterial two-hybrid system is able to detect interactions between small peptides and between entire proteins. It can also be used in genetic screening to identify proteins that interact with a specific target. In contrast to the yeast two-hybrid system, it can be used to study protein–protein interactions in the cytosol, as well as in the inner membrane. 

## 5. Pertussis Toxin

In addition to ACT, *B. pertussis* also produces PTX. Unlike ACT, which is also produced by *B. parapertussis* and *B. bronchiseptica*, PTX is exclusively produced by *B. pertussis*, although the *ptx* genes are present in the genomes of two other *Bordetella* species [69]. This toxin is composed of five different subunits, named S1 to S5 according to their decreasing molecular weights, arranged in a hexameric structure with a 1S1:1S2:1S3:2S4:1S5 stoichiometry (for review, see [70]). It is a member of the A-B toxin family, in which the A subunit (here S1) expresses enzyme activity and the B oligomers (here S2 to S5) are responsible for target cell receptor binding. Unlike ACT, which can enter the cell directly through the plasma membrane, PTX is taken up by receptor-mediated endocytosis and reaches the endoplasmic reticulum through retrograde transport, where the S1 is translocated into the cytosol. In the cytosol, S1 catalyses the transfer of the ADP–ribosyl moiety of NAD onto the alpha subunit of signal-transducing Gi/o proteins, which is the basis of PTX toxicity and the mechanism of hallmark features of pertussis, such as leukocytosis. 

### 5.1. Structure–Function Relationship of PTX

Before joining the Institut Pasteur de Lille, I was fortunate enough to isolate and sequence the structural gene of PTX [71]. This work established that the five subunits are produced as independent polypeptides encoded by a single polycistronic operon in the order of *ptxABDEC*, coding for S1, S2, S4, S5 and S3, respectively. Each polypeptide is synthesized with a typical cleavable signal peptide, suggesting that they are transported via the Sec apparatus through the inner membrane of *B. pertussis* into the periplasm, where the assembly into the holotoxin occurs. The S1 subunit contains regions of homology to the enzymatically active A subunits of cholera toxin and *E. coli* heat-labile toxin, two other ADP-ribosylating toxins, consistent with S1 being the catalytic subunit. S2 and S3 share a relatively high degree of sequence similarities. Each one is associated with an S4 subunit. 

Subsequent work of my group has shown that the S2/S4 and S3/S4 dimers display a certain degree of specificity in receptor binding [72]. While the S2/S4 dimer binds to haptoglobin, the S3/S4 dimer binds to the toxin receptor on the surface of Chinese hamster ovary cells. Deletions of Asn-105 in S2 and of Lys-105 in S3 reduced haptoglobin and Chinese hamster ovary binding, respectively. When both residues were deleted, PTX-mediated mitogenesis, an ADP-ribosylation-independent activity of the toxin, was abolished. When the S2 or the S3 gene was deleted from the *B. pertussis* chromosome, PTX analogues were produced that contain either two S3 or two S2 subunits [73], indicating that the two subunits can substitute each other, although this does not occur in wild-type *B. pertussis*. Interestingly, the toxin analogues lacking the S2 subunit were less efficiently secreted by *B. pertussis* than the wild-type version, while the deletion of the S3 subunit significantly increased PTX secretion, suggesting that S2 plays a role in the secretion of the toxin. Consistent with differences in receptor recognition by S2 and S3, the toxin analogue containing two S3 subunits was approximately 10 times more efficient in in vivo ADP-ribosylation of Gi proteins in Chinese hamster ovary cells, while the analogue containing two S2 subunits was roughly 100-fold less efficient than the natural toxin. When THP-1 cells were used in the in vivo ADP ribosylation assay, both mutant toxins were less efficient than the wild-type molecule. 

The five subunits all contain intrachain disulphide bonds: one in S1, three in S2 and S3 and two in S4 and S5. With Rudy Antoine, we have shown that the disulphide bond in S1 linking Cys-41 to Cys-200 is essential for assembly of S1 with the B oligomer [74]. Removal of each one of the two cysteines of S1 resulted in the secretion of the B oligomer without S1. 

### 5.2. Mechanism of PTX S1 Subunit Enzyme Activity

Once within the cytoplasm of the target cell, the disulphide bond of S1 must be reduced for ADP ribosylation to occur. Deletion of Cys-41 or of the adjacent Ser-30 abolished enzyme activity, while substitutions of Cys-41 by serine or glycine still allowed for detectable activity, albeit at reduced levels [75]. This allowed us to compare enzyme kinetics between the wild-type S1 and its mutant derivative, which indicated that the catalytic rate was not affected by the amino acid substitution but that the *K*_m_ of the mutant protein for NAD was increased relative to that of the wild-type enzyme, suggesting that Cys-41 is located close to the NAD-binding site. Of note, Cys-41 is located between two segments of S1 with strong sequence similarities to the A subunits of cholera toxin and *E. coli* heat-labile toxin. 

In addition to the S1 disulphide bond, the 27 C-terminal residues of this subunit are also essential for toxin assembly and secretion [74]. The 47 C-terminal residues of S1 are also involved in the enzyme activity, probably via binding to the Gi/oα protein acceptor substrate. Deletion of this region strongly reduced ADP ribosylation of the G protein, whereas the NAD glycohydrolase activity measured in the absence of the acceptor substrate was not affected [75,76]. Additional amino acid residues important for both ADP ribosyltransferase and NAD glycohydrolase activities were Trp-26 and Glu-129 [77]. While Trp-26 appeared to be important for NAD binding, Glu-129 is directly involved in catalysis. Photoaffinity labelling of S1 resulted in binding of the nicotinamide moiety of NAD to Glu-129 [78]. Furthermore, kinetic studies of an S1 analogue in which Glu-129 was substituted by aspartate showed a reduction in the catalytic rate by more than two orders of magnitude, while the apparent *K*_m_ value for NAD was not affected [79]. In addition, fluorescent quenching experiments conducted using increasing concentrations of NAD showed similar dissociation constants between the Glu-129 substitution mutant and wild-type S1, confirming that the substitution did not affect NAD binding in an important way. Importantly, when Glu-129 was replaced by cysteine, a disulphide was formed between Cys-41 and Cys-129, indicating that Glu-129 is located in proximity of the NAD-binding site in the tertiary structure of the protein. 

Another residue involved in catalysis is His-35 [80]. Substitutions of this residue resulted in a substantial reduction in *k*_cat_ values without strongly affecting the *K*_m_ value for NAD in an NAD glycohydrolase reaction. Replacement of His-35 by glutamine resulted in a 50-fold decrease in specific activity, while its replacement by asparagine reduced the specific activity by more than 500 fold, indicating that the glutamine side chain can partially mimic the imidazole ring of histidine. Although glutamine can mimic some of the hydrogen-bonding capacity of the ε-N of histidine, it does not have the proton transfer capacity of histidine and can thus not act as a true base. We therefore proposed that His-35 may not be directly involved in proton abstraction but rather in polarization via hydrogen bonding of the acceptor substrate water in the NAD glycohydrolase reaction or the Gi/oα cysteine in the ADP ribosyltransferase reaction. This increases the nucleophilicity of the polarized acceptor substrates, which may then attack the cleavable *N*-glycosidic bond of NAD. 

These studies, together with contributions from other laboratories, have allowed us to propose an enzymatic mechanism for PTX S1 [81]. In this model, the C-terminal region of S1 is crucial for Gi/oα binding and positioning of its acceptor amino acid, Cys-347, within the catalytic site of the enzyme. Glu-129 may act on NAD by retrieving the ribose 2′-OH proton, which weakens the nicotinamide–ribosyl bond. His-35 increases the nucleophilicity of the cysteine (or water molecule in the NAD glycohydrolase reaction), which can then attack the weakened *N*-glycosidic bond. Arg-9 within the active site would be involved in productive NAD binding via interaction with the phosphate groups of NAD. Changing its side chain length by replacing Arg-9 with lysine abolishes both NAD glycohydrolase and ADP ribosyltransferase activities. 

### 5.3. Applications: Development of a Live Attenuated Pertussis Vaccine

The most important applications of studies on PTX have been in the field of novel pertussis vaccine development. Genetic detoxification of PTX by replacing Arg-9 and Glu-129 with lysine and glycine, respectively, has been instrumental for the development of a safe and efficacious acellular pertussis vaccine [82]. This vaccine has been used in Italy for several years and is now in use in Thailand. Recent studies have shown that genetically inactivated PTX induces substantially longer-lasting antibody responses than chemically inactivated PTX [83], although the latter is now used in most acellular pertussis vaccines available.

We used the knowledge gained in studies on PTX to develop a live attenuated pertussis vaccine designed for nasal delivery [84]. Considering that *B. pertussis* is a strictly mucosal pathogen and naturally only colonizes the respiratory tract and that its contagiousness is amongst the highest known for a respiratory pathogen [85], we considered that the induction of mucosal immunity may be the optimal strategy to control pertussis at a population level [86]. Mucosal immunity and protection against infection is best achieved by prior infection with *B. pertussis* [87]. We therefore designed a live attenuated *B. pertussis* vaccine strain in order to mimic natural infection as closely as possible without causing disease. This vaccine, named BPZE1, is deficient in the production of dermonecrotic toxin and tracheal cytotoxin and produces genetically inactivated PTX through the replacement of Arg-9 and Glu-129 in S1 with lysine and glycine, respectively [84].

It was shown to protect mice from both lung and nasal colonization by virulent *B. pertussis* and to induce potent systemic and mucosal immune responses [88]. Importantly, the mucosal IgA response induced by BPZE1 appeared to be essential for protection by the vaccine against nasal colonization by *B. pertussis*. Unlike that induced by most current acellular vaccines, protection induced by BPZE1 was long-lasting [89]. It also induced lung and nasal tissue-resident memory T (T_RM_) cells that produce IL-17 and/or IFN-γ, similar to what has been observed after infection with virulent *B. pertussis*. We have shown that the induction of these T_RM_ cells, especially the IL-17-producing T_RM_ cells, is inhibited by vaccination with acellular pertussis vaccines, which prolongs nasal carriage of *B. pertussis* in acellular pertussis-vaccinated mice [90]. The vaccine was shown to be safe, even in severely immunocompromised mice [91,92]. We also evaluated its safety and protective capacity in non-human primates and showed that at a dose of up to 10^10^ colony-forming units (CFU) it did not induce any sign of disease in baboons yet was highly protective against pertussis disease and colonization by a highly virulent clinical *B. pertussis* isolate [93].

The vaccine has entered clinical development, and a first phase 1, first-in-human study carried out in Sweden has documented that it did not cause any significant adverse events after a single ascending dose from 10^3^ over 10^5^ to 10^7^ CFU delivered as nasal drops, as compared to a placebo administration [94]. It was able to transiently colonize the human respiratory tract in a dose-dependent manner and to induce serum antibodies against PTX, as well as other *B. pertussis* antigens, such as filamentous haemagglutinin, pertactin and fimbriae. These antibody levels did not decline for at least 6 months after vaccination, when the study was terminated. However, even at the highest dose used in that study (10^7^ CFU), only 5 out of the 12 participants were colonized by BPZE1 and induced antibodies to *B. pertussis* antigens. However, those that were not colonized had high levels of pre-existing antibodies to *B. pertussis* antigens, although they had never been vaccinated with pertussis vaccines, as they were born at a time when Sweden had stopped pertussis vaccination. This observation suggests that these pre-existing antibodies may have been generated by prior silent *B. pertussis* infection and that this prior infection may have induced immunity, preventing BPZE1 vaccine take, a hypothesis consistent with the notion that *B. pertussis* infection prevents subsequent *B. pertussis* reinfection.

This hypothesis was confirmed by a second phase 1 study also carried out in Sweden [95]. In this phase 1b study, subjects with high pre-existing antibody levels against *B. pertussis* antigens were excluded, and the vaccine was administered at doses of 10^7^, 10^8^ or 10^9^ CFU. In this study, colonization by BPZE1 was detected in roughly 80% of the participants, including in 10 out of 12 subjects who had received 10^7^ CFU. This also led to strong seroconversion against the *B. pertussis* antigens, with the highest seroconversion rate found in the 10^9^ CFU group. Even in the highest-dose group, the safety profile was comparable to that of the placebo group. Therefore, 10^9^ CFU was considered the optimal human dose to be used in the next clinical trials. At this dose, BPZE1 also induced *B. pertussis*-specific T-cell and robust memory B-cell responses [96]. The vaccine was also found to induce opsonizing antibodies, stimulating reactive oxygen species production in neutrophils and exerting bactericidal actions that were superior to those induced by acellular pertussis vaccines.

BPZE1 formulations were optimized as a lyophilized drug product that is stable for at least 2 years at storage temperatures up to room temperature [97]. It can now be delivered by the use of a spray device after reconstitution. The lyophilized and reconstituted formulation applied using the spray device was found to be more immunogenic and equally safe in a phase 2a study compared to the previous formulation administered as nasal drops [98] and was therefore used in a phase 2b trial.

In this phase 2b randomized, double-blind, placebo-controlled trial involving 300 volunteers, the safety of BPZE1 was confirmed at the level of both local tolerability and systemic reactogenicity [99]. The induction of long-lived *B. pertussis*-specific serum antibody production was also confirmed. In addition, this study showed that BPZE1, unlike the acellular vaccine comparator, also induced robust mucosal anti-*B. pertussis* secretory IgA responses, and elevated secretory IgA responses persisted for up to 254 days, when the study was terminated. Most importantly, this phase 2b trial provided proof of concept that nasal BPZE1 vaccination may protect against subsequent *B. pertussis* infection, as for 90% of the participants who had received a first dose of BPZE1, no bacteria were detected when BPZE1 was administered a second time 85 days later as an attenuated challenge dose, compared to only 30% of those who had received the acellular pertussis vaccine instead of BPZE1 as the first vaccine. Furthermore, for the 10% of the BPZE1 recipient for whom BPZE1 bacterial infection could be detected after attenuated challenge, the bacterial loads were substantially lower and cleared much faster than for those that were infected by BPZE1 after acellular pertussis vaccination. BPZE1 also induced PTX-neutralizing serum antibodies and complement-dependent bactericidal antibodies. Interestingly, these antibodies were able to kill pertactin-producing *B. pertussis*, as well as pertactin-deficient *B. pertussis*. In contrast, sera induced by acellular pertussis vaccination were only able to kill pertactin-producing *B. pertussis*. Given that in many high-income countries, pertactin-deficient *B. pertussis* currently largely predominates, this observation may illustrate an important additional asset of BPZE1 in the fight against pertussis.

Further clinical studies are currently underway to test the protective potential of BPZE1 against colonization by virulent *B. pertussis* using a controlled human challenge model (ClinicalTrials.gov: NCT05461131) and to evaluate its performance in school-age children (ClinicalTrials.gov: NCT05116241).

## 6. Conclusions

Pertussis is essentially a toxin-mediated disease, and in addition to LOS, ACT and PTX are major *B. pertussis* virulence factors involved in the pathogenesis of the disease, often acting in synergy. Scientists at Pasteur Institutes have made major contributions to the understanding of toxin action at the molecular level, their structure–function relationship and their involvement in pathogenesis using murine models. In addition, they have allocated efforts and time to translate this knowledge into novel tools for the prevention or treatment of important human diseases. This is in line with the vision of the founder of these institutes, Louis Pasteur himself, and his immediate disciples, who have devoted their lives to the study of major human health problems using scientific approaches, which have often led to effective solutions to these issues and have helped to improve global health overall.

## Data Availability

This review article does not contain new data.

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
