# Peer review of "Pasteurian Contributions to the Study of Bordetella pertussis Toxins"

_toxins, 2023, doi:10.3390/toxins15030176_

Round 1
Reviewer 1 Report
This nicely written review is an unusual piece of scientific text, as it specifically focuses only on results obtained with Bordetella toxins exclusively at the laboratories of the Institut Pasteur network (e.g. institutes in Paris and Lille). Hence, it is not citing and taking into account the results and progress achieved on the discussed toxins by other teams elsewhere. This makes the text somewhat imbalanced and incomplete, which may be misleading for some readers. On the other hand, this manuscript is still a useful reading on the history of some important advances made to the field by Institut Pasteur researchers.
I thus only pinpoint below some noted spelling errors that need to be fixed and I make several comments/recommendations that would be good to consider as minor modifications to the text.
Here are my points:
l. 28 – the human adapted strains of B. parapertussis differ from the ovine pathogenic ones and it would be good to make this clear. Rephrasing suggestion -: Bordetella parapertussis is an ovine pathogen and its human-adapted lineage causes milder whooping cough infections in humans.
(Please, note that no physician can distinguish on the basis of whooping cough symptoms if the infection is due to B. pertussis or B. parapertussis)
p. 1, l. 29 – B. bronchiseptica only causes symptomatic infections in people with reduced levels of immunity, such as elderly, HIV patients, immunosuppressed patients etc. Try to make this clear in the text.
p. 2, l. 64 -non-proteinaceous … lipooligosaccharide; p. 2, l. 66 – Jean-Marc
p.2, l. 70 – polysaccharide; l. 72 – secretion;
p. 2, l. 82 – the correct denomination of the enzyme is … adenylyl cyclase…
p. 3, l. 88 - … replace “the teams” by the name of the brilliant student who did that discovery, namely - Philippe Glaser …
p. 3, l. 96 – leukotoxin
l. 123 – in vitro; l. 127 – in vitro
l. 136 – calmodulin binding for…(not on)
l. 206 – it is important here to say that Barry et al. (J Bacteriol. 1991 Jan;173(2):720-6. doi: 10.1128/jb.173.2.720-726.1991) reported at the Bordetella conference in 1990 on the existence of a cyaC gene in the cya locus that was overlooked by Glaser et al. because it is transcribed transcribed divergently from the cyaABCDE operon. The report of Barry et al. then enabled Sebo et al. [ref. 31] to construct plasmids for co-expression of the cyaA and cyaC genes and demonstrate clearly that it is only the CyaC protein that posttranslationally activates the CyaA protoxin. This accomplishment should be put in value, since the possibility to produce large amounts of active recombinant ACT in E. coli made all the subsequent burst of structure-function studies on the CyaA toxin at all possible. Before that breakthrough, only very low amounts of ACT could be obtained from Bordetella cells and it was very tedious and limiting to work with it. Without the accomplishment of Sebo et al. [ref. 31] the use of CyaA as antigen delivery tool and the leap of ACT research by several teams would not have been possible, or would develop much later.
L. 323 – according to work of N. Guiso - ref. 53 and subsequent hybridoma generated and patents filed, the protection induced by ACT lacking the RTX domain is due to neutralizing antibodies binding to the segment of the residues 373-400 of the AC domain that block the translocation of the AC domain into cells , e.g. act like the 3D1 mAb of Lee SJ, Gray MC, Guo L, Sebo P, Hewlett EL. Infect Immun. 1999 May;67(5):2090-5. doi: 10.1128/IAI.67.5.2090-2095.1999.
L. 340 / target cells were invaded (not infected).
l. 401- this reviewer politely disagrees that PTX is the most complex bacterial toxin known so far, as stated by the author(s). For example, the typhoid toxin of S. typhi resembles very much PTX and on top of the ADP-ribosylating S1 subunit, it also has yet another subunit exhibiting a CDT activity (Nature. 2013 Jul 18;499(7458):350-4. doi: 10.1038/nature12377). More complex is also the structure and mode of delivery into cells of some other bacterial toxins, e.g. the large tripartite ABC-type toxin complexes (Nature. 2013 Mar 28;495(7442):520-3. doi: 10.1038/nature11987)
l. 605 – philosophy – however, even this corrected term should better be replaced by another one, e.g. vision, concept, underlying idea, or so…
Author Response
This nicely written review is an unusual piece of scientific text, as it specifically focuses only on results obtained with Bordetella toxins exclusively at the laboratories of the Institut Pasteur network (e.g. institutes in Paris and Lille). Hence, it is not citing and taking into account the results and progress achieved on the discussed toxins by other teams elsewhere. This makes the text somewhat imbalanced and incomplete, which may be misleading for some readers. On the other hand, this manuscript is still a useful reading on the history of some important advances made to the field by Institut Pasteur researchers.
I thus only pinpoint below some noted spelling errors that need to be fixed and I make several comments/recommendations that would be good to consider as minor modifications to the text.
Here are my points:
- 28 – the human adapted strains of B. parapertussis differ from the ovine pathogenic ones and it would be good to make this clear. Rephrasing suggestion -: Bordetella parapertussis is an ovine pathogen and its human-adapted lineage causes milder whooping cough infections in humans.
(Please, note that no physician can distinguish on the basis of whooping cough symptoms if the infection is due to B. pertussis or B. parapertussis)
Response: This is a good point. In the revised version I have made this clear by slightly modifying the sentence the reviewer suggests. I hope this is acceptable.
- 1, l. 29 – B. bronchiseptica only causes symptomatic infections in people with reduced levels of immunity, such as elderly, HIV patients, immunosuppressed patients etc. Try to make this clear in the text.
Response; I have now added that this occurs in subjects with immune deficiencies.
- 2, l. 64 -non-proteinaceous … lipooligosaccharide ; p. 2, l. 66 – Jean-Marc
Response: these spelling errors have now been corrected
p.2, l. 70 – polysaccharide; l. 72 – secretion;
Response: these spelling errors have now been corrected
- 2, l. 82 – the correct denomination of the enzyme is … adenylyl cyclase…
Response: I have now used “adenylyl cyclase” throughout the manuscript.
- 3, l. 88 - … replace “the teams” by the name of the brilliant student who did that discovery, namely – Philippe Glaser …
Response: as requested I have now made this replacement.
- 3, l. 96 – leukotoxin
Response: this spelling errors has now been corrected
- 123 – in vitro; l. 127 – in vitro
Response: This has now been modified
- 136 – calmodulin binding for…(not on)
Response: this has now been corrected
- 206 – it is important here to say that Barry et al. (J Bacteriol. 1991 Jan;173(2):720-6. doi: 10.1128/jb.173.2.720-726.1991) reported at the Bordetella conference in 1990 on the existence of a cyaCgene in the cya locus that was overlooked by Glaser et al. because it is transcribed transcribed divergently from the cyaABCDE operon. The report of Barry et al. then enabled Sebo et al. [ref. 31] to construct plasmids for co-expression of the cyaAand cyaC genes and demonstrate clearly that it is only the CyaC protein that posttranslationally activates the CyaA protoxin. This accomplishment should be put in value, since the possibility to produce large amounts of active recombinant ACT in E. coli made all the subsequent burst of structure-function studies on the CyaA toxin at all possible. Before that breakthrough, only very low amounts of ACT could be obtained from Bordetella cells and it was very tedious and limiting to work with it. Without the accomplishment of Sebo et al. [ref. 31] the use of CyaA as antigen delivery tool and the leap of ACT research by several teams would not have been possible, or would develop much later.
Response: Although Barry et al. were not at a Pasteur Institute I have now acknowledged their discovery in the revised version.
- 323 – according to work of N. Guiso - ref. 53 and subsequent hybridoma generated and patents filed, the protection induced by ACT lacking the RTX domain is due to neutralizing antibodies binding to the segment of the residues 373-400 of the AC domain that block the translocation of the AC domain into cells , e.g. act like the 3D1 mAb of Lee SJ, Gray MC, Guo L, Sebo P, Hewlett EL. Infect Immun. 1999 May;67(5):2090-5. doi: 10.1128/IAI.67.5.2090-2095.1999.
Response: I have now modified this sentence accordingly
- 340 / target cells were invaded (not infected).
Response: this has now been modified
- 401- this reviewer politely disagrees that PTX is the most complex bacterial toxin known so far, as stated by the author(s). For example, the typhoid toxin of S. typhiresembles very much PTX and on top of the ADP-ribosylating S1 subunit, it also has yet another subunit exhibiting a CDT activity (Nature. 2013 Jul 18;499(7458):350-4. doi: 10.1038/nature12377). More complex is also the structure and mode of delivery into cells of some other bacterial toxins, e.g. the large tripartite ABC-type toxin complexes (Nature. 2013 Mar 28;495(7442):520-3. doi: 10.1038/nature11987)
Response: I agree with the reviewer and have removed this sentence in the revised version of the manuscript.
- 605 – philosophy – however, even this corrected term should better be replaced by another one, e.g. vision, concept, underlying idea, or so…
Response: I like the word “vision”; so I replaced “philosophy” by “vision”
Reviewer 2 Report
In this manuscript, the authors summarized research activities of researchers in Pasteur Institutes (i.e., Pasteurian) on the study of toxins (LOS, ACT and PT) in Bordetella pertussis. The manuscript is informative and clarified the findings and issues of B. pertussis toxins (but is somewhat biased toward the topic of BPZE1…). The text is well written itself, but I found several parts are hard to understand without illustration. Structure diagrams of each toxin would be help to readers’ understanding. A few minor comments are listed below.
Line 81: Please check the instruction of MDPI for usage of comma and period for decimal point. I prefer to describe it as “177.3 kDa” or just “~180 kDa”.
Lines 127-132: I feel that this part digresses from the topic of ACT biogenesis.
Line 264: The characters are garbled.
Lines 506-507: Please cite the reference publications.
Lines 591-592: Please cite the reference publications.
Author Response
In this manuscript, the authors summarized research activities of researchers in Pasteur Institutes (i.e., Pasteurian) on the study of toxins (LOS, ACT and PT) in Bordetella pertussis. The manuscript is informative and clarified the findings and issues of B. pertussis toxins (but is somewhat biased toward the topic of BPZE1…). The text is well written itself, but I found several parts are hard to understand without illustration. Structure diagrams of each toxin would be help to readers’ understanding. A few minor comments are listed below.
Line 81: Please check the instruction of MDPI for usage of comma and period for decimal point. I prefer to describe it as “177.3 kDa” or just “~180 kDa”.
Response: I have replaced the comma by a period in the revised version
Lines 127-132: I feel that this part digresses from the topic of ACT biogenesis.
Response: I agree with the reviewer and have now removed this sentence
Line 264: The characters are garbled.
Response: I have tried to change this in the revised version
Lines 506-507: Please cite the reference publications.
Response: I am not sure I understand this comment. Reference #81 (now 82) was already cited in the original version; unless the reviewer refers to another statement
Lines 591-592: Please cite the reference publications.
Response: I am not sure I understand this comment. Reference #98 (now 99) was already cited in the original version; unless the reviewer refers to another statement